# Folding kinetics of an entangled protein

**Leonardo Salicari**[1,2], **Marco Baiesi**[1,2], **Enzo Orlandini**[1,2], **Antonio Trovato**[1,2]*

**1** Department of Physics and Astronomy "G. Galilei", University of Padova, Padova, Italy, **2** National Institute of Nuclear Physics (INFN), Padova Section, Padova, Italy

* antonio.trovato@unipd.it

## Abstract

The possibility of the protein backbone adopting lasso-like entangled motifs has attracted increasing attention. After discovering the surprising abundance of natively entangled protein domain structures, it was shown that misfolded entangled subpopulations might become thermosensitive or escape the homeostasis network just after translation. To investigate the role of entanglement in shaping folding kinetics, we introduce a novel indicator and analyze simulations of a coarse-grained, structure-based model for two small single-domain proteins. The model recapitulates the well-known two-state folding mechanism of a non-entangled SH3 domain. However, despite its small size, a natively entangled antifreeze RD1 protein displays a rich refolding behavior, populating two distinct kinetic intermediates: a short-lived, entangled, near-unfolded state and a longer-lived, non-entangled, near-native state. The former directs refolding along a fast pathway, whereas the latter is a kinetic trap, consistently with known experimental evidence of two different characteristic times. Upon trapping, the natively entangled loop folds without being threaded by the N-terminal residues. After trapping, the native entangled structure emerges by either backtracking to the unfolded state or threading through the already formed but not yet entangled loop. Along the fast pathway, trapping does not occur because the native contacts at the closure of the lasso-like loop fold after those involved in the N-terminal thread, confirming previous predictions. Despite this, entanglement may appear already in unfolded configurations. Remarkably, a longer-lived, near-native intermediate, with non-native entanglement properties, recalls what was observed in cotranslational folding.

## Author summary

Recently, a surprisingly large fraction of protein structures was shown to host topologically entangled motifs, whereby one protein chain portion is lassoed by a second portion, that loops between two residues in non-covalent contact with each other. Moreover, there is growing evidence that failure in adopting the correct entangled motifs may produce misfolded structures with impaired biological functions. Such structures are otherwise similar to the correct ones and can escape the cell quality control system for protein expression, leading to soluble and less functional protein species. Here, we study in detail the folding kinetics of an entangled small anti-freeze protein, using a simplified representation of the protein chain. We find a very rich folding behavior, unusual for small

**Data Availability Statement:** The whole dataset of MD trajectories, together with the analysis scripts used in the present work, can be accessed at https://researchdata.cab.unipd.it/984/.

**Funding:** AT was supported by EU funding within the MUR PNRR "National Center for HPC, BIG

DATA AND QUANTUM COMPUTING" (Project no. CN00000013 CN1). The funders had no role in study design, data collection and analysis, decision to publish, or preparation of the manuscript.

**Competing interests:** The authors have declared that no competing interests exist.

proteins, with different folding pathways. A fast pathway is followed if a crucial set of contacts is formed before lassoing takes place. If not, a misfolded structure which acts as a kinetic trap is formed, slowing down folding; in such structure, most of the contacts are correctly in place yet the lasso is not formed. The detailed understanding that we provide for a small protein may pave the way for similar studies for larger entangled proteins.

## Introduction

The biological functionalities of proteins are determined by the properties of their native states. The well defined compact native structure of a globular protein is achieved, in the aqueous cytoplasm medium, through the physical folding of the protein chain after assembly and release at the ribosome, where several chaperons assist the folding of the nascent chain. According to Anfinsen's thermodynamic hypothesis [1], small globular proteins are able to fold "in vitro" into the correct native structure in a reproducible manner, without the help of any cellular machinery. Despite the daunting complexity of the protein folding process, e.g. the important role played by the solvent degrees of freedom, a large body of research activity showed in recent decades that its kinetic and thermodynamic properties are encoded in simple descriptors of the native state structure, even within a coarse-grained representation [2–4].

For instance, the list of residue pairs that are in close spatial contact with each other in the native structure, the contact map [5–7], defines the corresponding native interaction network. Together with features characterizing the local native geometry of the protein chain, the contact map can be used to define an implicit solvent structure-based energy function, whose global minimum is attained for the native structure. This simple Go-like approach can be set up in several different flavours [8], and typically allows to predict the folding nucleus, i.e. the subset of residues whose interaction network needs to be formed for folding to proceed correctly, in good agreement with experimental data [9]. Notably, the same approach allows to predict successfully folding mechanisms more involved with respect to the typical two-state scenario, detecting the presence of thermodynamic and/or kinetic intermediates [9]. Coarse-grained structure-based models proved very insightful also in the study of the cotranslational folding process by means of numerical simulations [10].

Similarly, the loops formed between residues in contact with each other in the native structure have an average chemical length, the contact order [11], which is strongly correlated with the folding time for two-state folders [12–14]. The contact order is just one basic feature of the interaction network of native contacts. In general, the simpler the network, the faster the predicted folding [15]. The organization of the network of contacts, however, does not necessarily capture the topology of the protein backbone described as a curve in the three-dimensional space, as well as the possible formation of knots and other entangled motifs. The discovery of knots in a few proteins [16] was a surprise because they were believed to make the folding process unnecessarily hard. In fact, it was later realized that folding into knotted topologies could be surprisingly efficient, especially "in vivo" [17]. Their presence could be related to some biological function or stability requirement [18, 19], and the mechanisms allowing the folding dynamics to thread the protein backbone to form knots are under careful investigation [20–24]. Although knots could severely restrict the available folding pathways [18, 19], it is not obvious how proteins avoid the ensuing kinetic traps and fold into the topologically correct state [25–27]. In particular, the issue of at which stage of the folding process is the knotted topology formed spurred an intense debate [24, 28]. Experimental evidence shows that no general rule exists, since the knotted topology can be already present in the denatured state as for

the $\alpha/\beta$-knot methyltransferases YibK and YbeA [29], whereas the knot can form late along the folding pathway for the UCH-L1 protein with a $5_2$-knotted topology [30] or for the shallow trefoil knot of MJ0366 [31].

Recently, it was realized that motifs other than knots may lead, in some proteins, to three-dimensional structures with non trivial spatial arrangements. These include knotoids [32], slipknots [33], lassos [34, 35], pokes [36] and non-covalent lassos detected by Gaussian entanglement, as originally proposed by some of us [37–41]. Gaussian entanglement is related to the mathematical concept of linking number [42], being quantified by means of suitable generalizations of Gauss integrals for discretized and possibly open curves [43–45]. A crucial question is whether and how these topologically entangled motifs affect the protein energy landscape and the folding process [46].

Very recently, a striking connection was established between the presence of misfolded protein sub-populations during and just after protein translation and their entanglement properties. For example, an abundance of entangled motifs characterizes a subset of proteins prone to misfolding and aggregation under heat stress when newly synthesized, but not once matured [47]. This may suggest that those proteins rely more on the protein homeostasis machinery to reach their native states. Along the same lines, using coarse-grained structure-based models of protein translation, it was predicted that one-third of proteins can misfold into soluble less-functional states, that bypass the protein homeostasis network, avoiding aggregation and rapid degradation [48]. Such misfolded species were characterized as long-lived kinetic traps, native-like in several respects, with their misfolding due to non-native entanglement properties. Moreover, the shift in the competition between differently entangled misfolded subpopulations, along cotranslational folding pathways, was shown to determine the functional impact of synonymous mutations [49]. Interestingly, different metastable states were found to be populated, within a similar scenario of folding heterogeneity, also in a computational study of a multiply connected multidomain protein with interwoven chain topology [50].

The Gaussian entanglement was originally introduced to describe structurally entangled dimers [37, 51]. It was later found to be significantly correlated with the "in vitro" folding rate [38, 52], so that a higher degree of entanglement slows down the folding process. Interestingly, the Gaussian entanglement and the contact order can be combined to improve the prediction of folding rates [38], showing that the 3d topology adds some extra information over the underlying interaction network. Entangled loops are looped lasso-like segments of a protein chain displaying large Gaussian entanglement when threaded by another segment, possibly not looped, of the same protein chain. Recently, it was discovered that they are present in roughly one third of known protein domain structures [39], much more than knots [18]. Entangled loops can also be found, remarkably, in one fifth of trans-membrane protein domains [41]. Moreover, the amino acids at the end of a loop interact with each other with an energy significantly weaker, on average, if the loop is entangled [39].

According to the well established paradigm of minimal frustration [53, 54], energetic interactions in proteins are optimized in order to avoid as much as possible the presence of unfavorable interactions in the native state. Although non optimized interactions may result in kinetic traps along the folding pathway, some amount of residual frustration has been detected and related to functionality and allosteric transitions [55]. The above observation shows that non optimized interactions are present also in relation to topologically entangled motifs, an example of residual topological frustration. As a possible mechanistic explanation for this, it was hypothesized that the premature formation of entangled loops could cause the subsequent threading by another segment of the protein to become a kinetic bottleneck [39]. Therefore, it is likely better for entangled loops to be established in the later stages of the folding process. This hypothesis was further corroborated by another observation: entangled loops are found

in an asymmetric location with respect to the chain portion they are entangled with, the thread, such that the latter is found more frequently on the N-terminal side of the loop [39]. In the context of co-translational folding [56], this implies that entangled loops are synthesized at the ribosome, and hence folded, on average later than the thread. This "late entanglement avoids kinetic traps" hypothesis was verified within a toy lattice model for protein evolution targeting the folding speed [40]. Protein sequences optimized to fold as quickly as possible into an entangled native structure were indeed characterized by weak interactions at the ends of entangled loops.

In this contribution, we study the folding behaviour of the small antifreeze RD1 protein, which is natively entangled, by means of molecular dynamics simulation within a coarse-grained structure-based implicit solvent approach. Our aim is to verify, in general, how the presence of entangled motifs may affect the folding mechanism and, in particular, to test the "late entanglement" hypothesis. We also study the folding behaviour of the SH3 domain, as a natively non-entangled benchmark with a similarly small number of around 60 residues, whose "in vitro" folding properties are well characterized experimentally. The SH3 domain is known to display a simple two state folding kinetics, a feature reproduced by computer simulations employing different Go-like approaches [9, 57–59]. To avoid dealing with the possible peculiar role of disulphide bridges, no cysteines are present in the two proteins selected for our study.

## Results

### Go-like energy function

We use a simplified coarse-grained Go-like energy function to study the folding process by means of Langevin molecular dynamics. The energy function is similar to the one introduced in [9], based on a $C_\alpha$ representation of the protein chain (see the Structure-based energy function subsection for details). In brief, the native values of pseudo-bond lengths, pseudo-bond angles, and pseudo-dihedral angles are favored by either elastic quadratic terms (bond lengths/angles) or sinusoidal terms (dihedrals) in exactly the same way as in [9]. We model the long range pairwise interactions through a Lennard-Jones 12/6 potential, whereas a 12/10 Lennard-Jones potential was used instead in [9]. Attractive long range interactions are considered only for residue pairs found in contact with each other in the native structure. We define native contacts based on heavy atom positions (see the Structure-based energy function subsection for details), although in a different way from [9]. The weights of the different terms in the energy function are the same as in [9], being uniform across all residues. We implement Langevin thermostatted molecular dynamics through the LAMMPS software [60] (see the Langevin dynamics subsection for details). In this contribution we will present results from two different sets of Langevin molecular dynamics simulations: equilibrium-sampling ones at/around the folding transition temperature and refolding simulations below the folding transition temperature.

### Topological properties of the SH3 domain and the RD1 protein

In this work we study and compare the folding processes of two short protein chains with similar lengths, the SH3 domain and the antifreeze protein RD1. The cartoon structures of both proteins are shown in the upper panels of Fig 1. The histograms of the Gaussian entanglement values $G'$ for the loops joining the residue pairs in contact with each other in the native structure are shown in the lower panels of Fig 1 for both proteins (see the Gaussian entanglement subsection for how Gaussian entanglement is defined for a pairwise contact within a protein chain). In the rest of this work we will evaluate the overall entanglement of a protein

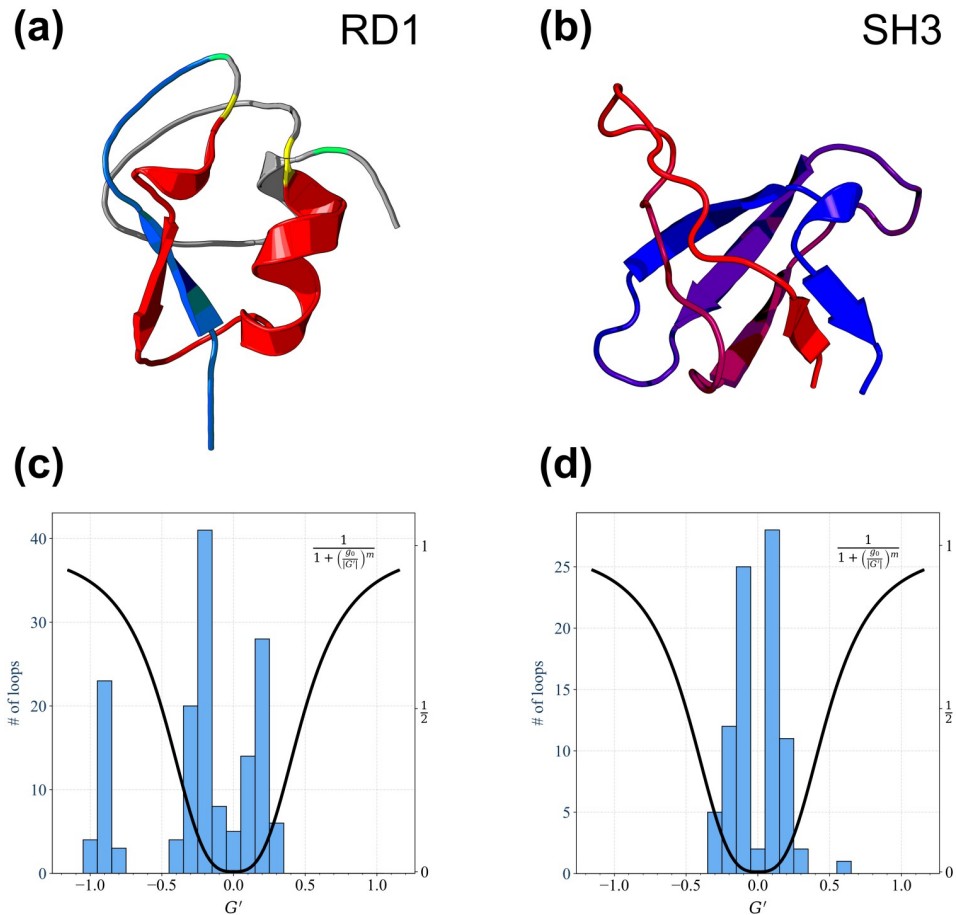

**Fig 1. Cartoon native structures and corresponding Gaussian entanglement histograms of the proteins studied in this work.** (a) Cartoon native structure of the Type III antifreeze protein RD1 (Protein Data Bank (PDB) code 1ucs), which exhibits a mixed $\alpha/\beta$ structure. To illustrate the native topological complexity of RD1, we show in red one entangled loop ($G' = -0.97$) between the contacting residues L17 and M41, in yellow, with the associated threading fragment in blue, the first 14 N-terminal residues (see the Gaussian entanglement subsection for the definition of entangled loops and threads). Whilst, the most entangled contact ($G' = -1.03$) is shown in green, between residues N14 and K61. (b) Cartoon native structure of the SH3 domain (PDB code 1srl), which has a mainly-$\beta$ structure, with five $\beta$ strands. (c) Histogram of the Gaussian entanglement values for the native loops in the RD1 protein. The unnormalized weight function used to evaluate the entanglement indicator $\langle G' \rangle = -0.68$ for the whole RD1 protein native structure is also shown. (d) Histogram of the Gaussian entanglement values for the native loops in the SH3 domain. The unnormalized weight function used to evaluate the entanglement indicator $\langle G' \rangle = 0.08$ for the whole SH3 domain native structure is also shown. The equation defining the unnormalized weight function is reported in the legends of both panels c and d. It is a Hill activation function with threshold $g_0 = 0.5$ and cooperativity index $m = 3$. The weight function needs to be properly normalized to compute the weighted average $\langle G' \rangle$. While we use the same unnormalized weight function in all cases, the normalization is specific to each distinct protein configuration (see the Gaussian entanglement subsection for details).

configuration by using a weighted average $\langle G' \rangle$ of the Gaussian entanglements for all the native contacts which are formed in that configuration. The weight function used in the average is shown in both lower panels of Fig 1. It is defined through a soft activation threshold $g_0 = 0.5$ in such a way that the more their values of $|G'|$ are above (respectively below) threshold, the more (respectively the less) the native contacts are likely to be entangled and the more (respectively the less) they contribute to the average $\langle G' \rangle$ (see the Gaussian entanglement subsection for details). Note that the entanglement can appear with either positive or negative chirality, depending on the sign of $G'$ or $\langle G' \rangle$.

src SH3 is a 56-residue fragment from the tyrosine protein kinase domain, well characterized in the literature as a prototypical two-state folder both experimentally [61] and in computer simulations [9, 57–59]. As seen in the lower right panel of Fig 1, only one native loop exhibits a Gaussian entanglement not much above the activation threshold, with $G' = 0.63 > g_0$. Hence, the weighted average indicator yields $\langle G' \rangle = 0.08$, showing that the SH3 domain does not exhibit topological complexity.

RD1 is a 64-residue long, Type III antifreeze protein from the Antarctic eelpout. The sub-microsecond protein folding kinetics of RD1 was studied using a photolabile caging strategy with time-resolved photoacoustic calorimetry [62]. As seen in the lower left panel of Fig 1, 30 different RD1 native loops exhibit a Gaussian entanglement well above the activation threshold, with $G' < -0.75$. As a result, the weighted average indicator yields $\langle G' \rangle = -0.68$, showing that RD1 is characterized by topological complexity. The largest Gaussian entanglement $G' = -1.03$ is associated to the native contact between residues N14 and K61 (in green in the upper left panel of Fig 1). The loop joining the two residues is entangled with the thread spanning the residues from N1 to I11. The set of the other 29 entangled native contacts, with $-1 < G' < -0.75$, is formed between residue pairs in the sequence portions P12-A24 and A34-L55. All these loops can be grouped together according to the clustering procedure performed in [39]. For illustration purposes, one such loop is highlighted in red in the upper left panel of Fig 1, with the contacting residues at its ends in yellow, and the corresponding thread in blue. Importantly, the thread associated with this cluster of entangled loops always consists of the first N-terminal residues, in a number ranging from 11 to 16. Therefore, all loops with $G' < -0.75$ entangle with essentially the same thread, a convenient illustration of the native topological complexity of the RD1 protein.

## Model benchmark and validation: folding thermodynamics of the non-entangled two-state folder

The results of 8 simulation runs at equilibrium (sample time series are shown in Fig A in S1 Appendix for the fraction of native contacts $Q$, the entanglement indicator $\langle G' \rangle$ and the RMSD from the native structure), carried out at different temperatures for the SH3 domain, were analyzed using the Weighted Histogram Analysis Method [63–65] (WHAM) in order to estimate the configurational entropy $S(Q)$ as a function of the fraction of native contacts $Q$. The configurational entropy can then be used to obtain the free energy profile $F(Q)$ as a function of $Q$, as well as other thermodynamic quantities such as the specific heat (see the Weighted histogram method subsection for details).

As shown in the right panel of Fig B in S1 Appendix, the specific heat exhibits a sharp peak as a function of temperature, whose position is used to locate the folding temperature $T_f$. The dimensionless free energy profile $F(Q)$ at the folding temperature is shown in the inset of the left panel of Fig 2, with the distinctive shape of two degenerate minima separated by a barrier, as expected for a two-state folding mechanism. The free energy profile can be used to define the transition state ensemble as the set of configurations sampled around the top of the free energy barrier (see the Ensemble definition and pathway classification subsection for details). The left panel of Fig 2 then shows the probability of native contact formation in the transition state ensemble.

In the left panel of Fig 2 we report the results obtained when the 12/6 Lennard-Jones potential is used for native pairwise attractive interactions in the energy function (see the Structure-based energy function subsection). The height of the free energy barrier between the low $Q$ unfolded state and the high $Q$ folded state is $\Delta F \simeq 0.25 \kappa_B T$, a much lower value than the one reported in [9], where a 12/10 Lennard-Jones potential and a different definition of the contact

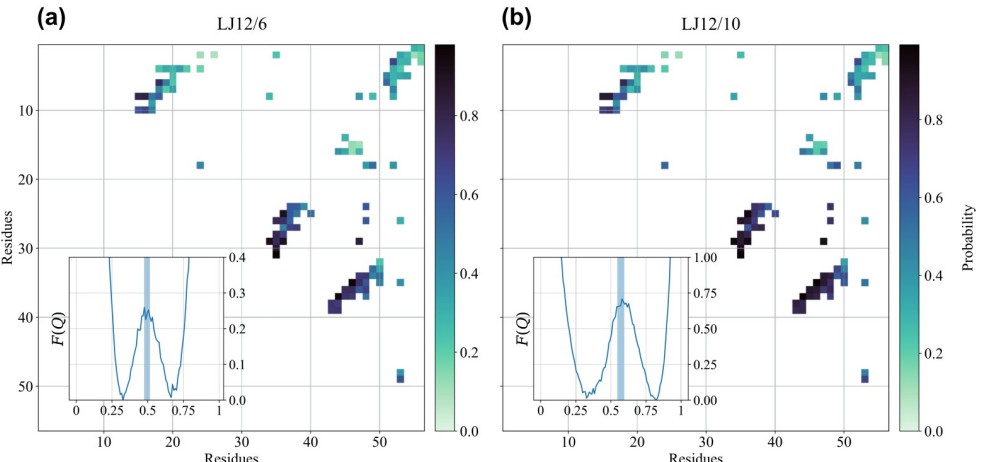

**Fig 2. Folding thermodynamics of the non-entangled SH3 domain.** Probability of native contact formation in the transition state ensemble at the folding temperature for the non-entangled SH3 domain. The inset shows the dimensionless free energy profile $F(Q)$ as a function of the fraction of native contacts $Q$ at the folding temperature. The shaded area in the inset highlights the interval of $Q$ values used to define the transition state ensemble. (a) Energy function with 12/6 Lennard-Jones potential. (b) Energy function with 12/10 Lennard-Jones potential. The similarity of the two contact maps can be quantified by computing correlation coefficients between the respective contact formation probabilities. Pearson's linear correlation coefficient: $r = 0.981$; Spearman's rank correlation coefficient: $r = 0.985$; in both cases $p$–value $< 10^{-77}$.

map are used instead, as discussed in the Go-like energy function subsection. The low height of the free energy barrier is also apparent in the small degree of cooperativity visible in the transitions between the folded and the unfolded states (see Fig A in S1 Appendix), with intermediate $Q$ values sampled rather frequently.

A higher free energy barrier, $\Delta F \simeq 0.70 \kappa_B T$, is obtained if we use the 12/10 Lennard-Jones potential in our energy function, as shown in the inset of the right panel of Fig 2. The increased cooperativity of the 12/10 flavour of the Lennard-Jones potential is likely to be a general property due to its faster decay at long distances. The shape of the free energy profile is now more similar to the one reported in [9] for the SH3 domain, although the height of the barrier is still lower and the positions that we find on the $Q$ axis for the unfolded and the folded states are less separated. These differences are then due to the dissimilar definition of the contact map that we use.

Crucially, however, the probability of native contact formation in the transition state is essentially not affected by the change in the flavour of the Lennard-Jones potential (the comparison between the two panels of Fig 2 is quantified in Fig 2 caption), being also similar to what found in [9] and in a more fine-grained structure-based model, with two interaction centers per residue [66]. The interactions between the 3 $\beta$-strands at the centre of the sequence are already partially formed, while the contacts involving the 20 N-terminal residues contribute less to the transition state structure, a description in agreement with experimental results [61]. The properties of the transition state ensemble are much more robust than the shape of the free energy profile when changes in the definition of native pairwise attractive interactions are made within a structure-based energy function.

In the rest of this work we will focus on features of the folding process similar to the properties of the transition state ensemble, such as the presence and nature of intermediate states; we will thus present results obtained by using the 12/6 Lennard-Jones potential, for the sake of computational efficiency.

## The two-state behaviour of the non-entangled SH3 domain does not depend on the entanglement indicator and does not change in the refolding kinetics

The middle panel of Fig A in S1 Appendix shows that the transitions between the folded and the unfolded states are hardly distinguishable for the SH3 domain when considering the entanglement indicator $\langle G' \rangle$. To confirm this, we used the latter as a second reaction coordinate, beside the fraction of native contacts $Q$. We evaluated the dimensionless free energy surface $F(Q, \langle G' \rangle)$ at the folding temperature as a function of both reaction coordinates, based on the data sampled in 8 long simulations at equilibrium all run at $T = T_f$. The related bidimensional contour plot, shown in the left panel of Fig C in S1 Appendix, confirms that the entanglement indicator $\langle G' \rangle$ is essentially irrelevant in the description of the two-state folding behaviour of the SH3 domain. On the other hand, regions in the unfolded state ($0.3 \lesssim Q \lesssim 0.4$), characterized by values of the entanglement indicator $|\langle G' \rangle| \sim 0.5$ close to the activation threshold $g_0$, are sampled at the $\sim 5\kappa_B T$ level for both chiralities. Relatively frequent fluctuations to these values can indeed be observed in the time series (see the middle panel of Fig A in S1 Appendix).

To check whether the folding behaviour of the SH3 domain possibly depends on temperature, 100 refolding kinetics trajectories were simulated at $T = 0.9 T_f$ for $4.17 \cdot 10^4$ MD time steps, with unfolded initial conditions (see the Ensemble definition and pathway classification subsection for details on the generation of unfolded initial conditions). The refolding temperature is selected to mimic physiological conditions, although it is important to emphasize that a map between implicit solvent simulations and real units is not trivial. In all 100 trajectories, the SH3 domain achieved refolding in the allotted simulation time, displaying the same two-state behaviour already observed at the folding temperature. Sample time series are shown in Fig D in S1 Appendix, from which we observe again that the entanglement indicator $\langle G' \rangle$ is not relevant for describing the refolding of the SH3 domain.

These observations are further confirmed using the entanglement indicator $\langle G' \rangle$ and the fraction of native contacts $Q$ as reaction coordinates. In the right panel of Fig C in S1 Appendix, we show the histogram contour plot, in log scale, in the $(Q, \langle G' \rangle)$ plane, obtained by grouping together the data from all refolding trajectories. Note that the right panel of Fig C in S1 Appendix, at variance with the left panel, does *not* represent a two-dimensional free energy surface, since it is obtained from non-equilibrium refolding trajectories in which the unfolding state is populated only transiently. Once this is taken into account, the comparison between the two panels of Fig C in S1 Appendix shows that the two-state nature of the folding mechanism of the SH3 domain does not change with temperature. The relatively frequent fluctuations of the entanglement indicator to values close to the activation threshold $g_0$ observed for both chiralities in the unfolded state do not depend on temperature as well.

## The entangled RD1 protein exhibits a two-state behaviour at the folding transition

The results of 8 long simulation runs at equilibrium (sample time series are shown in Fig E in S1 Appendix), carried out at different temperatures for the RD1 protein, were analyzed using the WHAM method [63–65] in order to estimate the configurational entropy $S(Q)$ as a function of the fraction of native contacts $Q$, allowing to obtain the free energy profile $F(Q)$ as a function of $Q$ and the specific heat (see the Weighted histogram method subsection for details).

As shown in the left panel of Fig B in S1 Appendix, the specific heat exhibits a sharp peak as a function of temperature, whose position is used to locate the folding temperature $T_f$. The

dimensionless free energy profile $F(Q)$ at the folding temperature is shown in the bottom horizontal inset of Fig 3, with the distinctive shape of a two-state folding mechanism, with two degenerate minima separated by a barrier.

The height of the free energy barrier between the unfolded state and the folded one is $\Delta F \simeq 3.2\kappa_B T$, a much higher value than the one found for the SH3 domain when using the 12/6 Lennard-Jones potential (see the Model benchmark and validation: folding thermodynamics of the non-entangled two-state folder subsection). This reflects the much more cooperative folding/unfolding transitions observed for the RD1 protein in Fig E in S1 Appendix, with intermediate $Q$ values sampled much less frequently. Within our coarse-grained structure-based approach, when comparing proteins of similar length, the presence of topological complexity is associated with a sharp increase in the cooperativity of the folding process.

To assess when the native entanglement is formed in the folding transition process for the RD1 protein, we used the entanglement indicator $\langle G' \rangle$ as a second reaction coordinate, beside the fraction of native contacts $Q$. We evaluated the dimensionless free energy surface $F(Q, \langle G' \rangle)$ at the folding temperature as a function of both reaction coordinates, based on the data sampled

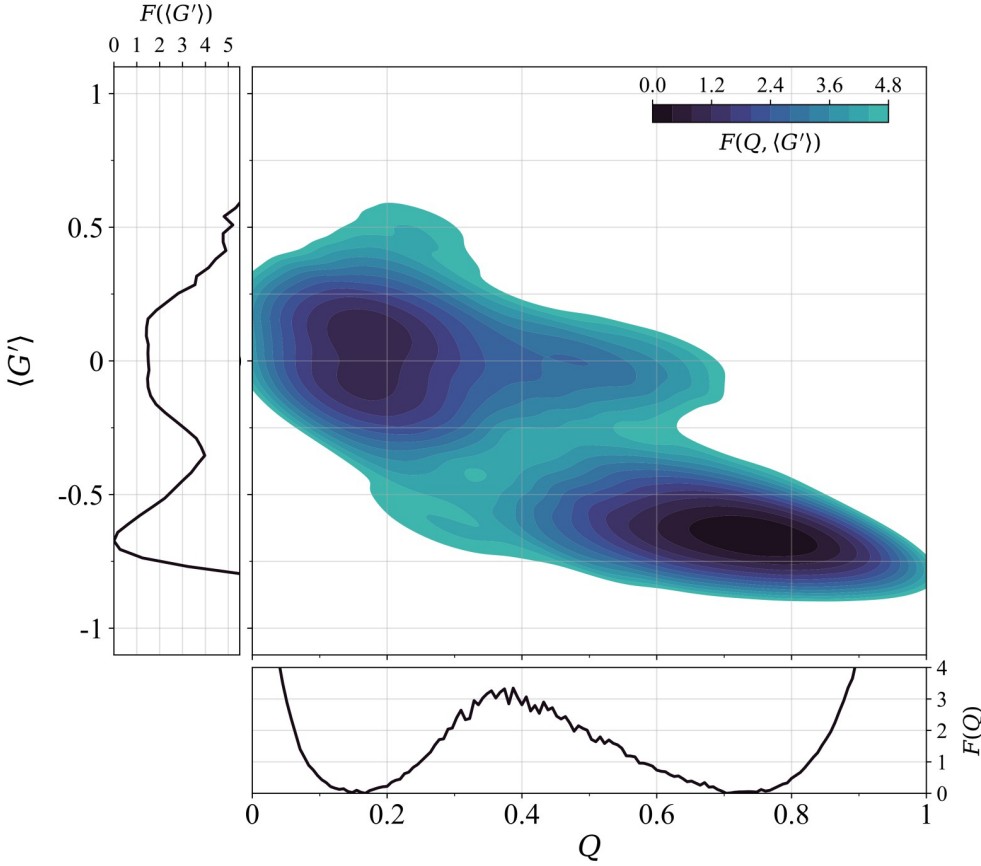

**Fig 3. Folding thermodynamics of the entangled RD1 protein at the folding transition temperature.** Contour plot of the dimensionless free energy surface in the $(Q, \langle G' \rangle)$ plane for the entangled RD1 protein. The free energy is evaluated as the negative of the log-scale histogram of data collected from 8 long equilibrium trajectories at $T = T_f$. Histogram negative log-counts are smoothed using KDE (see the Ensemble definition and pathway classification subsection for details) and shifted in order for their minimum to be 0. Contour levels correspond to approximately 0.4 in log-histogram units. The bottom horizontal inset shows the dimensionless free energy profile $F(Q)$ obtained using the WHAM method (see the Weighted histogram method subsection). The left vertical inset shows the dimensionless free energy profile $F(\langle G' \rangle)$ after the projection onto the entanglement indicator.

in 8 long simulations at equilibrium, all run at $T = T_f$. The related bidimensional contour plot, shown in Fig 3, confirms that the RD1 protein is a two-state folder at the folding transition. At the same time, it illustrates how the entanglement reaction coordinate is crucial for a proper description of the folding process of the RD1 protein. The projection of the free energy surface onto the entanglement indicator $F(\langle G' \rangle)$ is shown in the left vertical inset of Fig 3, highlighting that the native entanglement needs to be at least partially formed ($|\langle G' \rangle| \lesssim g_0$) in the transition state at the top of the free energy barrier. The diffuse nature of the transition state, however, can only be appreciated in the full bidimensional contour plot, with the height of the barrier estimated as $\Delta F \simeq 3.2 \kappa_B T$ ($Q \sim 0.4$, $\langle G' \rangle \sim -0.3$). Interestingly, experimental results show that transition states are represented by similarly broad free energy barriers, for both folding pathways of the UCH-L1 protein with a $5_2$-knotted topology [67]. One can also observe unfolded configurations ($Q \simeq 0.2$) populated at the $5 \kappa_B T$ level, that can be entangled with either the native or the opposite chirality ($|\langle G' \rangle| \simeq 0.6$).

## The entangled RD1 protein exhibits a longer-lived non-entangled intermediate which acts as a kinetic trap when refolding below the folding transition temperature

To check whether the folding behaviour of the RD1 protein possibly depends on temperature, 100 refolding trajectories were simulated at $T = 0.9 T_f$ for $8.33 \cdot 10^4$ MD time steps, with unfolded initial conditions (see the Ensemble definition and pathway classification subsection for details on the generation of unfolded initial conditions). To compare the simulations with the refolding ones for SH3, the same ratio between refolding temperature and $T_f$ is used. Note that the allotted simulation time is twice the one used for refolding trajectories of the SH3 domain. Interestingly, only 81 trajectories refolded correctly to the native state, exhibiting a variety of different folding pathways, as exemplified in the sample time series shown in Fig 4. A "fast" cooperative refolding, similar to the folding transition taking place at $T = T_f$, was observed in 52 trajectories (see the left panel of Fig 4 and S1 Video). Other 29 trajectories displayed the presence of a kinetic intermediate that delays successful refolding, acting as a trap (see the middle and the right panels of Fig 4). The remaining 19 trajectories non-achieving successful refolding within the allotted simulation time appear to be trapped in the same intermediate at the end of the simulation. The trap intermediate (IT in the following) is not entangled ($\langle G' \rangle \simeq 0$), whereas its fraction of native contacts ($Q \simeq 0.6$) is closer to the folded state ($Q \simeq 0.8$) than to the unfolded one ($Q \simeq 0.2$).

After trapping in IT, successful refolding trajectories can reach the native state through a direct "threading" transition (see the middle panel of Fig 4 and S2 Video, where one can appreciate how the N-terminal chain portion is able to thread through an already formed loop), or fold through the fast channel after "backtracking" to the unfolded state at low $Q$ (see the right panel of Fig 4). In a few trajectories we observe multiple transitions from the unfolded state to IT, before eventually folding through either direct threading or backtracking to the fast channel.

The different types of observed refolding trajectories are summarized in Table 1 for the RD1 protein, together with the corresponding mean folding times. The distributions of folding times for each type of refolding trajectories are shown in Fig F in S1 Appendix, together with the refolding time distribution for the SH3 domain. The presence of both backtracking and threading implies that IT is neither on-pathway nor off-pathway, with a complex mechanism emerging for the refolding of the RD1 protein. The IT intermediate does indeed act as a kinetic trap since the mean folding time is $8 \div 10$ times longer for any pathway going through it with respect to the fast folding channel. Similar complex mechanisms, with a two-state behaviour at

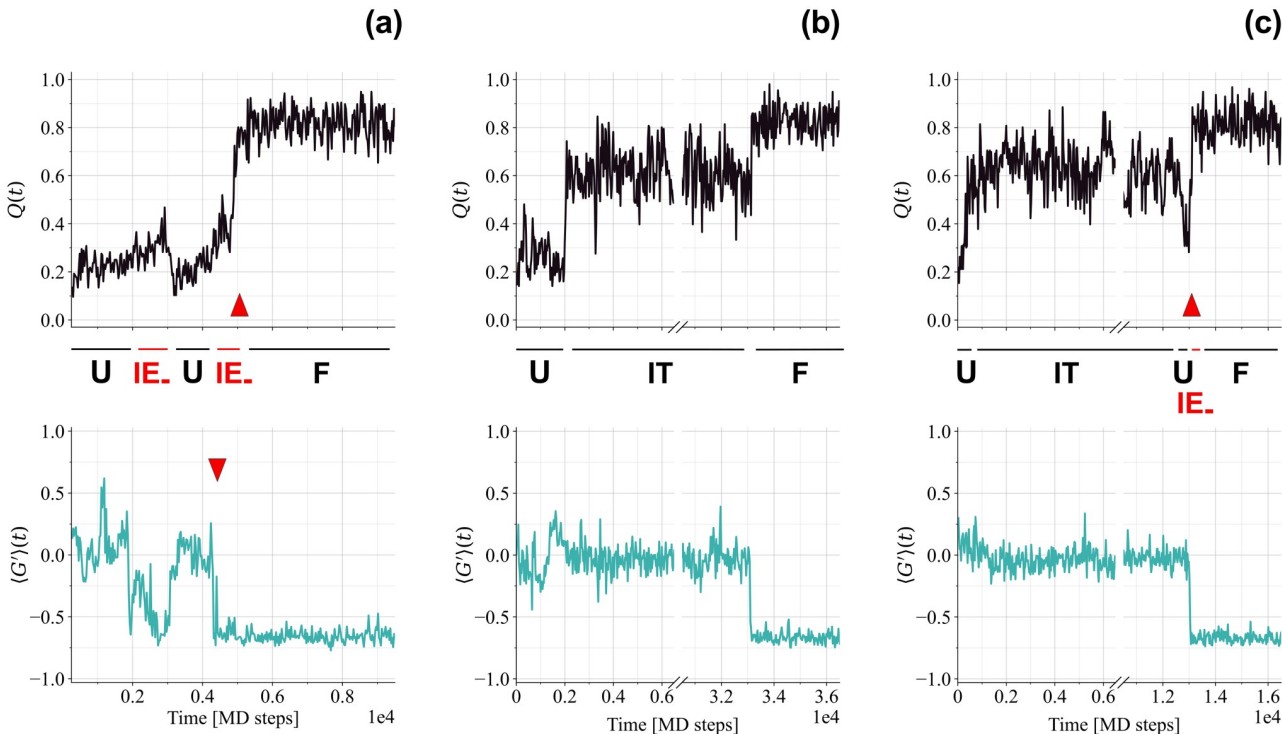

**Fig 4. Time series from refolding trajectories for the RD1 protein.** Upper row: fraction of native contacts $Q$ with labels referring to the visited states. Lower row: entanglement indicator $\langle G' \rangle$. (a) Example of "fast" folding trajectory. The two red arrows illustrate how in the folding transition the entanglement indicator reaches native values before the fraction of native contacts does the same, delimiting the short visit to the IE_ intermediate prior to folding. In this trajectory, it is shown an example of a visit to IE_ followed by a return to the unfolded state. (b) Example of "threading" trajectory. (c) Example of "backtracking" trajectory. The red arrow marks the final folding transition.

equilibrium and a kinetic intermediate appearing below the folding transition temperature, are known to occur for other proteins [68]. Interestingly, the folding time distribution for the RD1 protein through the fast folding channel is similar to the one observed for the SH3 domain.

**Table 1. Different refolding pathways for the entangled RD1 protein.**

| Folding pathway | | Number of instances | Mean folding time |
|---|---|---|---|
| Fast Folding | $\cdots \rightarrow IE_- \rightarrow F$ | 52 | $0.46 \times 10^4$ |
| Threading | $\cdots \rightarrow IT \rightarrow F$ | 25 | $3.68 \times 10^4$ |
| Backtracking | $\cdots \rightarrow IT \rightarrow U \rightarrow IE_- \rightarrow F$ | 4 | $4.77 \times 10^4$ |
| Trapped in Intermediate | $\cdots \rightarrow IT$ | 19 | $> 8.33 \times 10^4$ |

The number of times a given folding pathway type is observed in the refolding of the RD1 protein is reported, together with the average folding time observed for each subset of trajectories. Horizontal dots represent possible transitions among meta-stable states, whereas tilted dots stress that trajectories classified in the fast folding pathway do not explore the IT ensemble. The folding time of a given trajectory is defined as the first time for which $Q \geq 0.75$ and $\langle G' \rangle \leq -0.5$ for RD1 and $Q \geq 0.7$ for SH3 (see the Exponential fit of contact formation curves subsection for details). Time is measured in MD steps (see the Langevin dynamics subsection for details).

## The entangled RD1 protein exhibits a short-lived entangled intermediate which guides refolding through the fast channel

A better characterization of the different folding pathways observed for the RD1 protein can be gained by using the entanglement indicator $\langle G' \rangle$ as a second reaction coordinate besides the fraction of native contacts $Q$, as already done for the SH3 domain. In Fig 5 we show the histogram contour plot in the $(Q, \langle G' \rangle)$ plane, in log scale, obtained by grouping together the data from all 100 refolding trajectories. Unlike Fig 3, this contour plot does *not* represent a two-dimensional free energy surface, since it is obtained from non-equilibrium refolding trajectories, in which the unfolding state and the intermediate states are populated only transiently. Conversely, the folded state is the only non-transient one, so the depth of the corresponding minimum in Fig 5 depends on the total simulation time of refolding trajectories. On the other hand, the depths of the unfolded and intermediate minima depend on their characteristic lifetimes and on how frequently they are reached in the refolding trajectories. The different folding pathways observed in our simulations and discussed in the previous subsection can be traced using the arrows shown in Fig 5, which reveals several interesting features.

The presence of a populated intermediate state, characterized by the *absence* of the native topological complexity ($\langle G' \rangle \simeq 0$) and by a large fraction of native contacts already formed ($Q \simeq 0.6 \div 0.65$), is apparent. This is the kinetic trap IT which slows down the folding of the RD1 protein, as already discussed in the previous subsection. The folding pathway can then proceed directly to the native state F through the threading channel, or go back to the unfolded state U.

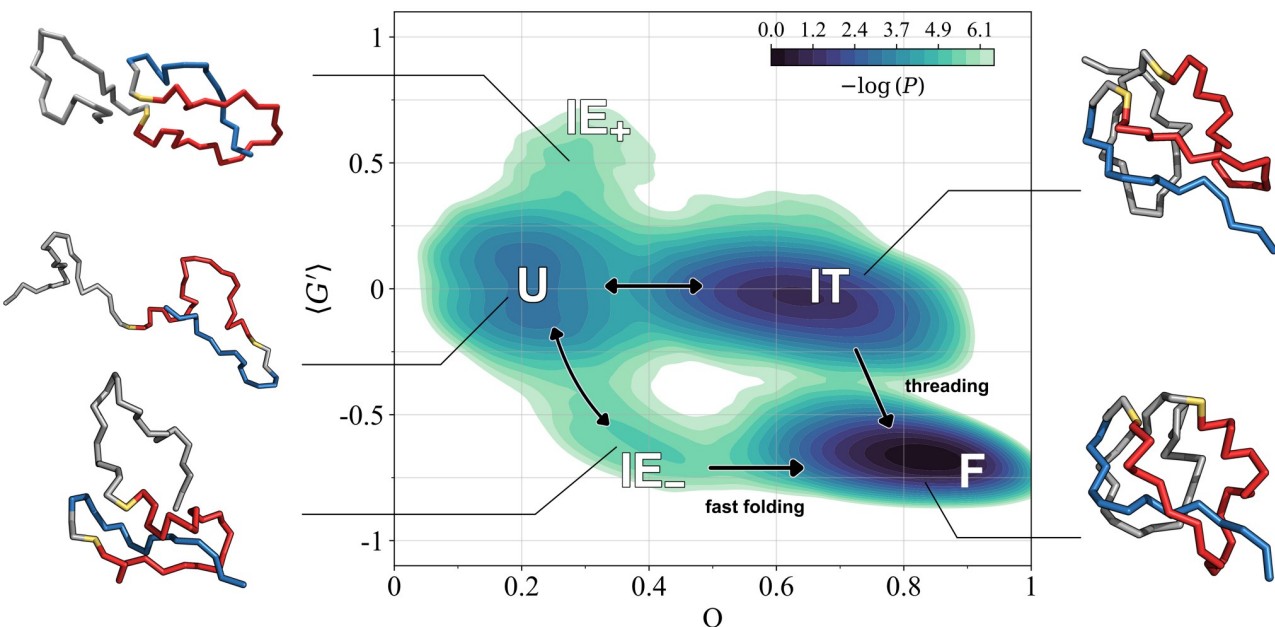

**Fig 5. Folding pathways of the entangled RD1 protein below the folding transition temperature.** Log-scale histogram contour plot in the $(Q, \langle G' \rangle)$ plane from 100 refolding trajectories for the RD1 protein at $T = 0.9T_f$. Histogram negative log-counts are smoothed using KDE (see the Ensemble definition and pathway classification subsection for details) and shifted in order for their minimum to be 0. Contour levels and the colour scale are the same as in Fig 3. Letters refer to the unfolded (U), folded (F), trap intermediate (IT), entangled intermediate with negative chirality (IE$_-$) states, and to the positive chirality configurations populated during refolding (IE$_+$). Representative snapshots for each state are shown with the same colour as in the upper left panel of Fig 1. In IT, the non-correct threading of the N-terminal portion (in blue) through the loop (in red) which eventually becomes entangled in the folded state F is apparent. The arrows show the different transitions observed: fast folding from U to F through IE$_-$, possible disentanglement from IE$_-$ to U, trapping from U to IT, folding by threading from IT to F, backtracking from IT to U.

Furthermore, fast folding to the native state appears instead to proceed by populating a short-lived intermediate (IE_ in the rest of the paper) where the native topological complexity, with a negative chirality, is basically already established ($\langle G' \rangle \simeq -0.65 \div -0.6$), whereas the fraction of native contacts is just moderately larger than in the unfolded state ($Q \simeq 0.35 \div 0.4$). An instance of a short passage through IE_ prior to folding along the fast channel can be seen in the left panel of Fig 4, delimited by the two red arrows. We also observe several trajectories in which the RD1 protein unfolds back to U, after reaching IE_, as for example in the left panel of Fig 4.

Interestingly, configurations (IE_+ in the rest of the paper) almost unfolded ($Q \simeq 0.3$), yet entangled with positive chirality ($\langle G' \rangle \simeq 0.6$), appear to be sampled at the 2.4 contour level with respect to U in log histogram units. IE_+ configurations appear to be a "mirror image" of the configurations in IE_, where the low number of native contacts that are formed makes it possible for entanglement to be established with an opposite chirality with respect to the native one.

The overall refolding behaviour at $T = 0.9T_f$ is quite different from the one observed at equilibrium at the folding transition (see Fig 3). No intermediate state is present at $T = T_f$, although the populated plateau extending from the unfolded state at $\langle G' \rangle \simeq 0.0$ towards higher $Q$ values is reminiscent of IT, and only a "diffuse" folding pathway is observed, with no clear separation between the different folding channels, instead evident in Fig 5. This establishes the kinetic nature of the intermediates populated in the refolding trajectories at $T = 0.9T_f$.

In Fig 5 we show also configuration snapshots representing the different ensembles. The entangled loop, the corresponding thread, and the color code are the same as in the upper left panel of Fig 1. The overall structure of the IT configuration is similar to the correctly folded F configuration; the only difference involves the N-terminal portion, which in IT is not threading the loop that eventually becomes entangled in the folded structure F. The threading is already achieved in both the IE_ and IE_+ configurations, although with opposite chirality in the latter, whereas the C-terminal part of the chain is essentially unfolded. The few contacts present in the U configuration are localized in the N-terminal part of the chain.

## The early formation of contacts involving the natively entangled thread is crucial to select the fast folding channel for the RD1 protein

In order to gain mechanistic insight into the difference between the fast (U → IE_ → F) and the slow (U → IT → ⋯ → F) refolding channels, we focus on the different properties of the intermediate states IE_ and IT.

In the previous subsection, we already discussed how the threading refolding channel involves the *N*-terminus to thread an already formed loop in order to establish the native topological complexity (see S2 Video and the configuration snapshots IT and F in Fig 5).

This can be confirmed in a more quantitative way, by looking at the contact formation probability in the IT configuration ensemble in the upper right part of Fig 6, where it is compared to the contact formation probability in the IE_ ensemble, in the lower left part. The ensembles were defined by collecting the configurations sampled in the corresponding local minima of the bi-dimensional contour plot, as shown in Fig G in S1 Appendix (see the Ensemble definition and pathway classification subsection for details).

Most of the native contacts are already formed in IT, with the exception of several contacts (see the red arrow in the upper right part of Fig 6) involving the 6 N-terminal residues. On the contrary, interactions involving the latter residues are among the most likely formed in the short-live intermediate IE_ that is encountered along the fast folding channel (see the red arrow in the lower left part of Fig 6). Note that almost all other contacts in IE_ are essentially

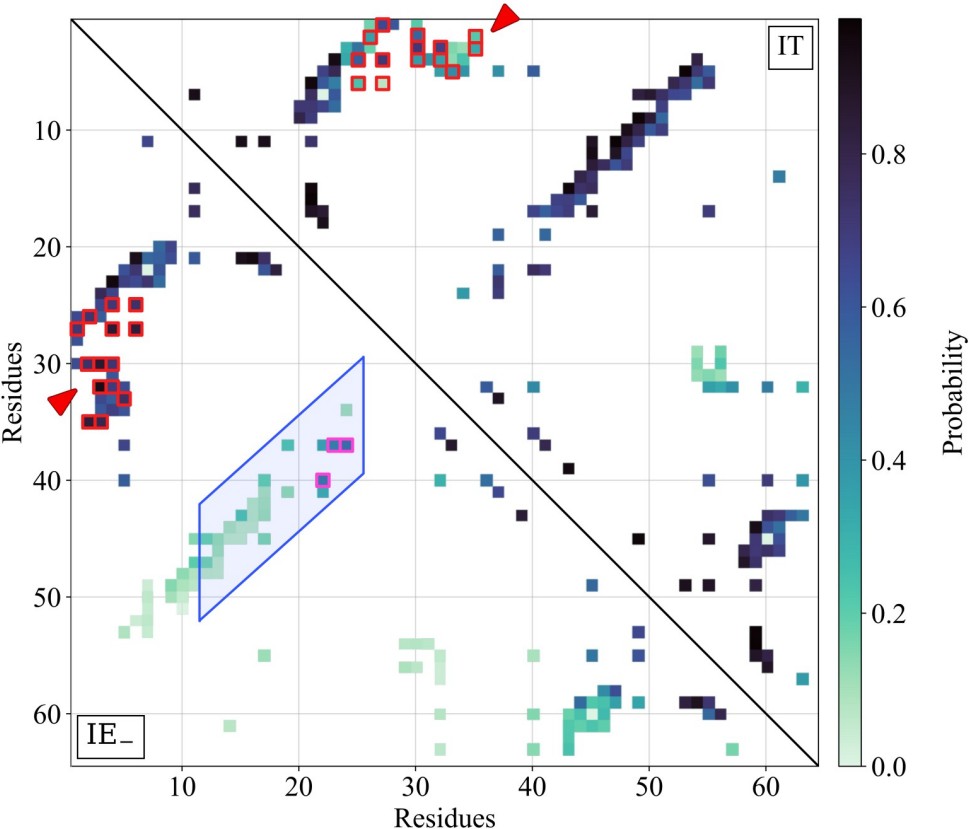

**Fig 6. Intermediate state ensemble contact maps for the entangled RD1 protein.** Probability of native contact formation for the entangled RD1 protein in the longer-lived IT ensemble (above diagonal) and the short-lived IE_ ensemble (below diagonal). Both intermediate state ensembles are detected at $T = 0.9T_f$ in Fig 5. The red arrows mark the location of the native contacts involving the first 6 N-terminal residues, which are likely to be formed in IE_ whereas they are less likely to be formed in IT. Among the former ones, the "trap-avoiding" native contacts framed in red are those whose formation probabilities differ most in the two ensembles (see the Ensemble definition and pathway classification subsection for details). The shaded area framed in blue contains 28 out of the 30 natively entangled contacts, with $G' < -0.75$. The native contacts framed in magenta are the "first-entangling" ones more likely to be formed in IE_ (see the Ensemble definition and pathway classification subsection for details).

not formed yet. The "trap-avoiding" native contacts framed in red in Fig 6 were selected as the ones whose formation probabilities differ most in the two ensembles, being more likely formed in IE_ (see the Ensemble definition and pathway classification for details).

It is also interesting to investigate whether any native contact is specifically originating the native-like entanglement observed in IE_. The natively entangled contacts, with $G' < -0.75$, are mostly contained in the shaded area framed in blue in the lower left part of Fig 6; almost all of them are essentially not formed yet. However, a few of them stand out because their formation probability is higher than the average one in IE_ (see the Ensemble definition and pathway classification subsection for details). These "first-entangling" contacts are framed in magenta in Fig 6, illustrating the origin of the entanglement observed in IE_ configurations.

Overall, this analysis provides evidence that, for efficient and fast folding of the RD1 protein to occur, the thread needs to fold correctly *prior* to the closure of the natively entangled loop. When this does not happen, the contacts that would be entangled in the correctly folded structure, form without originating the native entanglement, trapping the RD1 protein in the IT intermediate.

## Entangled contacts fold at the later stages along the fast folding channel of the RD1 protein

Using the 52 trajectories whereby the RD1 protein refolds at $T = 0.9T_f$ to the native state through the fast channel (see Fig 5), we wish to characterize the folding kinetics separately for each native contact. The average probability of contact formation (see the Exponential fit of contact formation curves subsection in Materials and Methods for details) increases with time upon refolding, from its initial values in the unfolded state U, $p_U$, to its final value in the folded ensemble F. As exemplified in Fig 7, the contact formation probability curves can be fit to an exponential kinetics with a characteristic rate $k$, which we call the contact folding rate. We report in S1 File the exponential fits for all the native contacts of the RD1 protein.

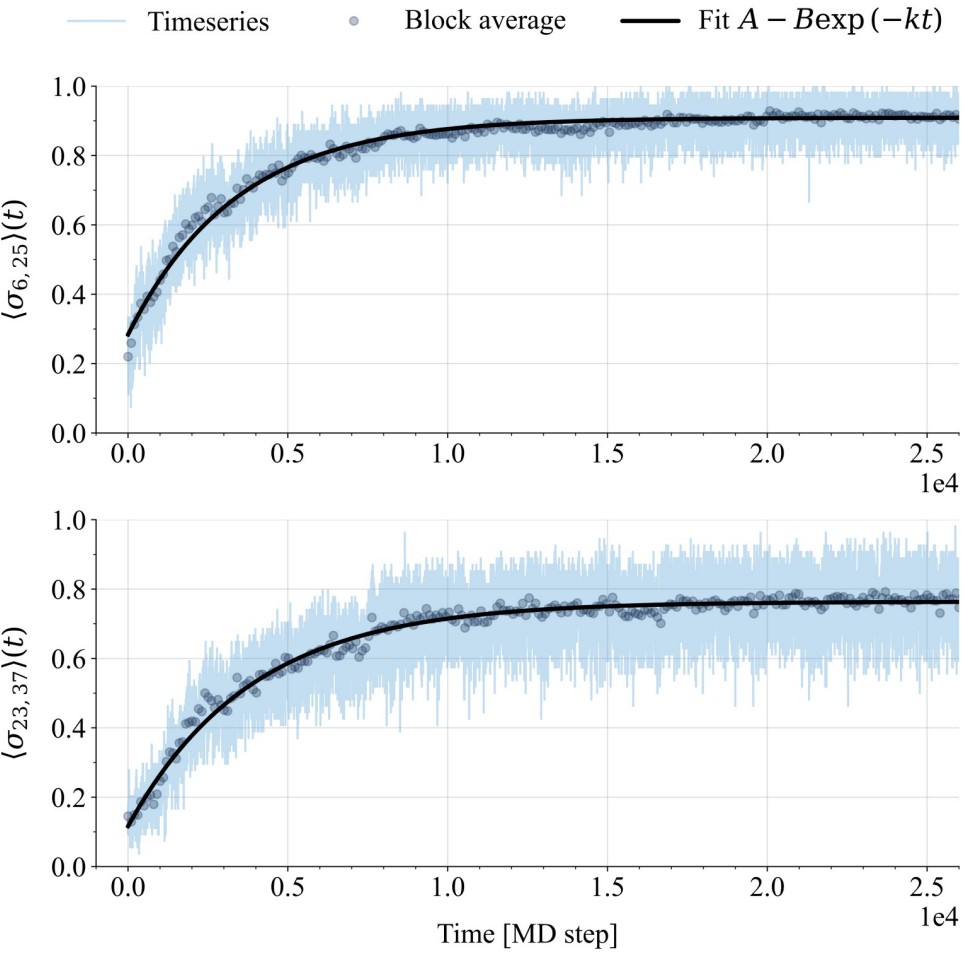

**Fig 7. Examples of exponential fits to the contact formation probabilities for the entangled RD1 protein.** Two examples of native contact formation probabilities as a function of time, averaged over the 52 refolding trajectories that fold to the RD1 protein native structure at $T = 0.9T_f$ through the fast channel, and of the corresponding exponential fits. Time is measured in MD steps. Both the average time series and the corresponding block averages (see the Exponential fit of contact formation curves subsection for details) are plotted (see legend). The block averages are fit to the exponential function reported in the legend and the resulting fits are shown in the plots. The fit parameters are $A$, the saturation value of the contact formation probability in the folded state, $B$, the gain of the former quantity in going from the unfolded to the final folded state, and $k$, the contact folding rate (See Exponential fit of contact formation curves section for details). Top row: native contact between V6 and E25; one of the "trap-avoiding" N-terminal thread contacts framed in red in Fig 6 and in the left panel of Fig 8. Bottom row: native contact between K23 and I37; one of the "first-entangling" contacts framed in magenta in Fig 6 and in the left panel of Fig 8.

The contact folding rates $k$ and the formation probabilities $p_U$ in the unfolded state U of the different native contacts of the RD1 protein are shown in the left panel of Fig 8, together with the corresponding $|G'|$ in the native structure. Notably, entangled contacts (with $|G'| > 0.75$) appear in general to be among the ones with the lowest values of both $k$ and $p_U$ (dark circles in the lower-left corner of the left panel of Fig 8).

In the left panel of Fig 8, it is interesting to identify the native contacts that were selected in Fig 6, based on the different properties of the intermediate state ensembles characterizing IT and IE$_-$. The "trap-avoiding" contacts, involving the N-terminal thread, are framed in red in the left panel of Fig 8 as well. The exponential fit for the contact formation probability of one of these contacts is shown in the top panel of Fig 7. The left panel of Fig 8 shows that any "trap-avoiding" contact folds at a significantly faster characteristic rate than any entangled contact. Notably, the $p_U$ feature also discriminate neatly the set of "trap-avoiding" contacts (with higher $p_U$ values) from the set of entangled contacts (with lower $p_U$ values).

The three contacts selected as "first-entangling" in Fig 6 are framed in magenta in the left panel of Fig 8 as well. The exponential fit for the contact formation probability of one of them is shown in the bottom panel of Fig 7. The left panel of Fig 8 shows that the "first-entangling" contacts can be discriminated against almost all other entangled contacts by their larger $p_U$ values.

Taken together, these results thoroughly confirm the hypothesis that, for a fast and efficient folding process of the RD1 protein, entangled contacts need to fold slower than the contacts involving the thread. On the other hand, provided trap-avoiding contacts are correctly folded,

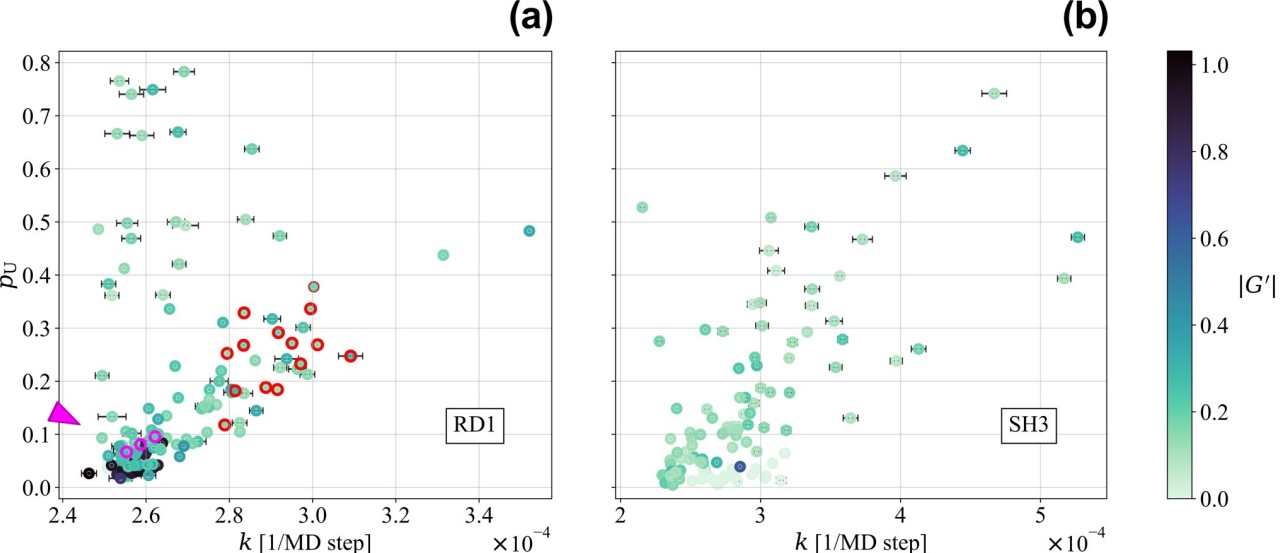

**Fig 8. Contact formation probabilities in the unfolded state and contact folding rates for both the entangled RD1 protein and the non-entangled SH3 domain.** The parameters $p_U$, the contact formation probability in the unfolded state U, and $k$, the characteristic rate at which this probability increases from $p_U$ to its final value in the folded ensemble F, are shown for each native contact. $p_U$ and its standard deviation are computed from 1500 unfolded configurations sampled with the same procedure used to select the initial configurations for refolding simulations (see Ensemble definition and pathway classification section for details). Contact folding rates $k$ are obtained through exponential fits of the contact formation probabilities as a function of time, averaged over different refolding trajectories (see Fig 7 for specific examples of such fits). When not shown, standard deviations are smaller than the marker size for both observables (see Exponential fit of contact formation curves section for details). The colour scale refers for each contact to the $|G'|$ of the corresponding loop. The darker the color the more entangled the loop. (a) RD1 protein. Only the 52 trajectories achieving refolding through the fast channel are considered in the average. The "trap-avoiding" and the "first-entangling" contacts identified in Fig 6 are framed in red and magenta, respectively. The position in the plot of the latter set is marked by the magenta arrow. (b) SH3 domain. All 100 refolding trajectories are included in the average.

fluctuations already present in the unfolded state allow "early-entangling" contacts to generate the native entanglement. Moreover, the coherence of the results highlighted in Fig 6 and in the left panel of Fig 8 validates "a posteriori" our choice of the entanglement indicator $\langle G' \rangle$. The selection of the "trap-avoiding" and "first-entangling" contacts in Fig 6 relies on the definition of the intermediate state ensembles, in which the use of $\langle G' \rangle$ as the second reaction coordinate in Fig 5 plays indeed a crucial role. On the other hand, the parameters shown in the left panel of Fig 8 can be obtained regardless of the definition of the intermediate state ensembles.

Finally, we analyzed the 100 refolding trajectories for the SH3 domain (in this case we observe correct folding to the native structure in all cases at $T = 0.9T_f$) in a similar way, by fitting the average probability of contact formation as a function of time with an exponential function for every single native contact. We report in S2 File the exponential fits for all the native contacts of the SH3 domain. The contact folding rate $k$ estimated from the fit and the contact formation probabilities $p_U$ in the unfolded state U, are shown in the right panel of Fig 8 for the SH3 domain, together with the $|G'|$ associated to the corresponding contact in the native structure. However, in the absence of entangled motifs, we observe no correlation between the Gaussian entanglement values and the contact parameters shown in the plot. Both panels of Fig 8 appear in general similar, with essentially the same scale for $p_U$ and $k$ in both proteins. This highlights that the entangled RD1 protein is able to fold, at least along the fast folding channel, as fast and efficiently as the non-entangled SH3 domain. The larger spread in $k$ values observed for the SH3 domain is consistent with the smaller degree of cooperativity already discussed previously (see Fig B in S1 Appendix).

## Discussion

Non-covalent lasso-like entanglement motifs were recently found in a large fraction of protein domain structures [39]. These motifs are present even in small proteins, such as the 64-residues Type III antifreeze RD1 protein studied in this work, as shown in the upper left panel of Fig 1. Entanglement can be quantified for each pairwise residue-residue contact, by its Gaussian entanglement value, $G'$, checking to what extent the loop joining the two contacting residues is threaded by other protein chain portions. In this work, we propose to use a novel single-valued descriptor of the topological entanglement for a protein configuration, computed as a weighted average $\langle G' \rangle$ over all contacts/loops formed in that configuration, as shown in the lower left panel of Fig 1 for the native structure of the RD1 protein.

To investigate how entangled motifs are formed and how the topological frustration due to entanglement is dealt with during the folding process, we adopted one of the earliest and most popular structure-based models [9], where the protein chain is coarse-grained at the level of one CA atom per residue, with two minor adjustments. We expect the obtained results not to change qualitatively with different definitions of the energy function or different coarse-graining schemes, as long as the topological complexity is properly described.

Our model validation on the non-entangled SH3 domain reproduced its well-known two-state folding behaviour (see the inset of the left panel of Fig 2), although with a much-reduced cooperativity of the folding transition with respect to the original model [9]. Crucially, however, the probability of native contact formation in the transition state ensemble is not affected by changing the flavour of the energy function (see Fig 2). We thus expect the results presented for the natively entangled RD1 protein, based on the presence and features of intermediate states, to be robust ones.

Notably, our simulation results for the refolding kinetics of the RD1 protein (see Fig 4) show that the notion of a misfolded longer-lived intermediate, resembling the native structure (high fraction of native contacts), but without the native entanglement, can be carried over

from large proteins (hundreds of residues) in the cotranslational context [48, 49] to the folding of an already fully synthesized small natively entangled protein. The longer-lived intermediate IT acts indeed as a kinetic trap, slowing down the folding time by one order of magnitude (see Table 1).

Moreover, we find that entangled unfolded configurations IE, with a low fraction of native contacts, are visited dynamically with relative ease, even with an opposite chirality (IE$_+$) with respect to the native RD1 structure (see Fig 5). Note that entangled motifs with flipped chiralities are observed for misfolded species of large proteins in structure-based coarse-grained simulations of cotranslational folding [48]. Such misfolded structures were predicted to be stable for at least hours by means of physics-based all atom simulations [69], suggesting that they are not an artifact due to using a structure-based energy function with an entangled native structure. More generally, our results highlight the possible role played by entangled yet essentially unfolded configurations in determining which folding pathway is chosen. Different entangled motifs could indeed originate already in the early stages of the folding process, with a low fraction of native contacts, to be then possibly "frozen" when the fraction of native contacts is increased. It will be interesting to study whether and how the kinetic partitioning between the many differently entangled misfolded subpopulations observed in cotranslational folding simulations [48, 49] can arise through differently entangled unfolded configurations/subpopulations. The entanglement parameter $\langle G' \rangle$ introduced here, should be an essential indicator for characterizing the partitioning of states.

The RD1 protein is not trapped forever in the longer-lived intermediate IT, at variance with what is observed in the context of cotranslational folding for much larger proteins [48, 49]. In the latter case, entangled motifs are typically deep (far away from sequence termini), whereas for the small RD1 protein the entanglement due to the threading of the N-terminal portion (see the upper left panel of Fig 1) is shallow. As a result, both the direct folding from IT to the native structure upon threading (see the middle panel of Fig 4 and S2 Video) and the backtracking from IT to the unfolded state (see the right panel of Fig 4) are observed. It will also be interesting to check for how long some longer proteins with deep entangled motifs can be trapped in mis-entangled compact intermediates, and whether backtracking is the only option for correct refolding. Very recently, the characteristic lifetime of mis-entangled compact intermediates was predicted to be of the order of days at room temperature for a large (283 residues) protein, by extrapolating all-atom simulations at higher temperatures [69].

The presence of a short-lived intermediate IE$_-$ that secures the formation of the native-like entanglement, and is crucial for fast and efficient folding without trapping, is particularly interesting. IE$_-$ is characterized by a low fraction of native contacts, in keeping with the general kinetic mechanism suggested above to generate entangled motifs. We expect the experimental observation of the short-lived entangled intermediate IE$_-$ to require a high temporal resolution. On the other hand, single-molecule force spectroscopy experiments might help in this respect, since the threading channel could become much less sampled upon unzipping. Magnetic tweezer experiments were recently shown to be useful in probing complex folding landscapes [70].

The correct early folding of the N-terminal thread, forming a set of "trap-avoiding" contacts, is crucial for fast and efficient folding (see Fig 6). On average, if the contacts that are entangled in the native state (blue shaded region in Fig 6) fold before the latter ones, trapping in IT without formation of the native-like entanglement follows. Notably, only a few specific "first-entangling" contacts are enough to originate the native-like entanglement in the IE$_-$ intermediate. This overall picture is fully confirmed by a different analysis, based on fitting to exponential curves the contact formation probabilities, averaged over the trajectories for which refolding occurs through the fast pathway, without trapping in IT (see Figs 7 and 8).

The coherency of the results obtained in Figs 6 and 8 confirms the validity of the entanglement indicator $\langle G' \rangle$ as a reaction coordinate.

Remarkably, the rich refolding behaviour observed for the natively entangled RD1 protein in Fig 5 is a kinetic effect, occurring below the folding transition temperature, at $T = 0.9 T_f$. On the other hand, a more typical (for small proteins) two-state folding behaviour is observed at equilibrium at $T = T_f$ (see Fig 3). The presence of kinetic transient intermediates, that are not populated at equilibrium, is well known in general in protein folding [68, 71], generating a rich and complex folding behaviour. However, this observation for a protein as small as RD1 is striking, pointing again to the crucial role of entanglement.

All the results discussed for the RD1 protein confirm the "late entanglement" hypothesis made on general grounds [39]; entangled contacts need to fold on average in the latter stages of a fast and efficient folding process and their premature folding (before the thread they entangle with in the native structure is already folded) leads to trapping into kinetic bottlenecks without formation of the native entanglement. Yet, thanks to few "first-entangling" contacts, this does not preclude the observation of the short-lived intermediate IE₋, conducive to the fast-folding of the RD1 protein. This mechanism may be reminiscent of what is found for some knotted proteins within numerical simulation of structure-based models, highlighting the presence of the knotted topology in the early stages of the folding process [72–74]. On the other hand, the role of non-native interactions, not considered in the model used here, should also be taken into account [75, 76].

An interesting question is whether and how the complex behaviour exhibited by the RD1 protein is modified for cotranslational folding. It is plausible that trapping in IT would be disfavoured; entanglement is characterized in RD1 by an N-terminal thread, that should then form more easily, in the cotranslational context, the crucial "trap-avoiding" contacts needed to direct the folding along the fast pathway through the entangled intermediate IE₋. Conversely, it will be interesting to study if different patterns of folding behaviour may emerge for entangled native structures characterized by a C-terminal thread. Longer entangled proteins may present different coexisting entangled motifs, and/or ones characterized by multiple windings ($|G'| \simeq 2$, $|G'| \simeq 3$) [39]; coarse-grained models such as the one we used here are clearly needed to study their folding mechanisms in details since additional reaction coordinates are required to properly describe the formation of multiple windings/motifs.

The few experimental results available for the "in vitro" folding of the RD1 protein are consistent with our findings. RD1 is a Type III antifreeze protein from the Antarctic eelpout, *Lycodichthys dearborni* [77]. The folding equilibrium upon chemical denaturation was shown to be a reversible two-state process with no populated intermediates, for the HPLC-12 type III antifreeze protein from the North-Atlantic ocean pout *Macrozoarces americanus* [78], a close isoform of the RD1 protein. The folding kinetics cannot be monitored by general detection methods, since RD1 lacks an intrinsic fluorescence probe and any clear spectroscopic differences between the folded and unfolded states [62]. However, a photolabile caging strategy followed by time-resolved photoacoustic calorimetry allowed to gather evidence of two distinct refolding events with different characteristic times [62]. Significantly, this result is consistent with our observation of the intermediate IT providing a kinetic bottleneck, since the fast event was shown to be associated to a large volume change (the fast folding from U to F in our interpretation), whereas the slow event to a much smaller volume change (the threading from IT to F in our interpretation). IT and F share in fact a similar compactness (see Fig 5).

The RD1 "antifreeze" protein domain can have other functions; it is found included in the multi-domain *E. coli* sialic acid synthase [79] and its deletion causes the loss of enzyme activity [80]. The antifreeze function is due to the presence of a flat ice-binding surface, which includes most of the residues in the V9-M21 chain portion and Q44 [77] and is highly conserved across

different isoforms [77, 81]. The N-terminal S4-A7 residues and several residues in the V26-E35 stretch are also highly conserved [77], although they do not participate in the ice-binding surface, consistently with the crucial role their "trap-avoiding" interactions play in the folding mechanisms highlighted by our simulations (see Fig 6). Interestingly, the I37 residue, which is involved in 2 of the 3 "first-entangling" contacts without being part of the ice-binding surface, is found to be conserved as well [77]. However, a more careful analysis should gauge the contribution to residue conservation from overall stability, since there is currently no evidence of residue conservation due to kinetic reasons for small non-entangled fast folding proteins [82].

The predictions obtained with our coarse-grained simulations need further confirmations by more refined models, as well as by experiments. Future studies would greatly benefit from the ability to predict and engineer RD1 mutants that either stabilize or abrogate the IT intermediate, as done with folding intermediates for other proteins [83]. More generally, as a cold-adapted protein with a specific antifreeze function, the RD1 protein provides a much more interesting subject for both molecular evolution and biotechnology processes [84]. We believe the presence of a non-covalent lasso entangled motif highlighted here for the RD1 protein, together with the prediction of its major impact on the folding mechanisms, could be of great interest in the above fields.

## Materials and methods

### Structure-based energy function

For our Molecular Dynamics (MD) simulations, we implemented an alpha carbon, coarse-grained model with a structure-based energy function. The functional form of the potential energy is based on the one used by Clementi and colleagues [9]. A protein composed of $N$ residues is modelled as a virtual polymer chain where each monomer is characterized by its alpha carbon atom. The vectors $\{\mathbf{r}^i\}_{i=1}^{N}$ are the residue position vectors, $\{\theta^i\}_{i=1}^{N-2}$ are the pseudo-bond angles of three subsequent residues, and $\{\phi^i\}_{i=1}^{N-3}$ are the pseudo-dihedral angles corresponding to four residues. The potential energy takes the form:

$$
\begin{aligned}
V \quad &= \sum_{i=1}^{N-1} \varepsilon_r^i (r^{i\,i+1} - r_0^i)^2 + \sum_{i=1}^{N-2} \varepsilon_\theta^i (\theta^i - \theta_0^i)^2 \\
&+ \sum_{i=1}^{N-3} \varepsilon_\phi^i \left\{ \left[ 1 - \cos(\phi^i - \phi_0^i) \right] + \frac{1}{2} \left[ 1 - \cos(3(\phi^i - \phi_0^i)) \right] \right\} \\
&+ \sum_{i+3<j}^{\mathrm{NAT}} 4\varepsilon_C^{ij} \left[ \left( \frac{\sigma^{ij}}{r^{ij}} \right)^{12} - \left( \frac{\sigma^{ij}}{r^{ij}} \right)^{6} \right] \\
&+ \sum_{i+1<j}^{\mathrm{NON}} V_{\mathrm{NN}}(r^{ij})
\end{aligned}
$$

where "0" subscripts specify the values that are extracted from the protein PDB structure.

The first term, where $r^{ii+1} = |\mathbf{r}^{i+1} - \mathbf{r}^i|$ is the virtual bond length, ensures the connectivity of the virtual chain, while the second and the third characterize its rigidity. For residue pairs in contact with each other in the native structure, the non-bonded interactions implicitly take into account the effective attraction induced by solvent effects through a Lennard-Jones 12/6 potential. $\sigma^{ij} = 2^{-1/6} r_0^{ij}$ is the corresponding zero-crossing point, where $r_0^{ij}$ is the native contact distance identified in the contact map analysis. A cut-off is imposed for $r^{ij} > 2.5\sigma^{ij}$ and no tail correction is applied. For residue pairs not in contact with each other in the native structure,

the repulsive-only term $V_{NN}(r^{ij})$ models excluded volume effects through a Lennard-Jones 12/6 potential, with parameters $\varepsilon_{NN}^{ij}$ and $\sigma_{NN} = 4$ Å, cut at $2^{1/6}\sigma_{NN}$ and shifted such that the potential is zero at the cut-off. The energy parameters are chosen to be uniform for all residues, $\varepsilon_r^i = 100\varepsilon$, $\varepsilon_\theta^i = 20\varepsilon$, $\varepsilon_\phi^i = \varepsilon_C^{ij} = \varepsilon_{NN}^{ij} = \varepsilon$, so that $\varepsilon$ sets the overall energy scale.

Consistently with previous work on entangled contacts [39], two residues are said to form a native contact in the coarse-grained description if, in the all-atom representation, any two non-Hydrogen atoms in different residues are closer than 4.5 Å. However, the same definition was used in other implementations of structure-based models [75]. The native contact length $r_0^{ij}$ is then set to the distance between the two alpha carbon atoms of the corresponding residues. Native contacts, and hence non-bonded attractive interactions, are considered only for residue pairs not involved in the same pseudo-bond or pseudo-dihedral angle. In MD simulations, a contact between residues $i$ and $j$ is formed if $r^{ij} < g\,r_0^{ij}$, where the choice of $g$ does not strongly affect thermodynamic and kinetic observables [85]. In the present work, $g = 1.2$.

## Langevin dynamics

We perform MD by simulating the Langevin equation of motion:

$$m_i \frac{d^2\mathbf{r}^i}{dt^2} = -\gamma \frac{d\mathbf{r}^i}{dt} - \nabla V + \mathbf{R}^i \tag{1}$$

where the mass $m_i = m\ \forall i$. On the right-hand side the force terms are: a drag force with drag coefficient $\gamma$, the conservative force $-\nabla V$ (see the Structure-based energy function subsection for details), and the Gaussian stochastic force $\mathbf{R}^i$, which satisfies $\langle R_k^i(t)R_l^j(t')\rangle = 2\kappa_B T\gamma\delta_{ij}\delta_{kl}\delta(t-t')$, where $T$ is the temperature of the heat bath. The drag and the random force mimic the interaction between the solvent and the system, consequently thermostatting the latter to the average temperature $T$. $\kappa_B = 1$ sets temperature units to be the same as energy. We use reduced units, rescaling mass by $m$ and energy by $\varepsilon$, whereas $a = 1$Å is used to rescale lengths. The time rescaling factor should then be $a\sqrt{m/\varepsilon}$; however, mapping to real-time units is not trivial and other possibilities have been proposed [86, 87]. In reduced time simulation units, $m/\gamma = 0.1$, and the integration time step is $\Delta t = 0.001$. In the present work, time is shown in plots and table entries as "MD steps", where 1 MD step corresponds to 24000 integrations of the Langevin equations.

The LAMMPS software [60] (3Mar2020 Version) is used to perform MD. The simulation of the above Langevin equations is implemented through the `fix langevin` option coupled with `fix nve`. The random number generator for the stochastic force is initialized for each trajectory in a different way. The integration scheme implemented is the velocity-Verlet. The whole dataset of MD trajectories, together with the analysis scripts used in the present work, can be viewed at https://researchdata.cab.unipd.it/984/.

## Weighted histogram method

The WHAM Method [63–65] uses histograms collecting system configurations, with respect to reaction coordinates, to compute thermodynamic observables. Here, we use an approximation that employs the fraction of native contacts $Q$ in place of the energy as a reaction coordinate. Although $Q$ is not a deterministic function of the energy, the minimally frustrated structure-based model reduces the energy fluctuations for configurations with the same value of $Q$, resulting in a good approximation [9]. $Q$ has the advantage of naturally binning histograms.

Let $R$ be the number of simulations at equilibrium with inverse temperatures $\beta_i$, $i = 1, \ldots, R$ ($\kappa_B = 1$). Let $\mathcal{N}_i(Q)$ be the number of configurations with a fraction of native contacts equal to

$Q$ in the $i$-th simulation, with $n_i = \sum_Q \mathcal{N}_i(Q)$ total entries. The configuration partition function in the microcanonical ensemble $W(Q)$ can be estimated as a linear combination of these histograms:

$$W(Q) = \sum_{i=1}^{R} p_i \frac{\mathcal{N}_i(Q)}{n_i} e^{\beta_i \langle E \rangle_Q - f_i} \ , \tag{2}$$

where $f_i = \beta_i F_i$ are dimensionless free energies and $p_i$ are normalized weights to be determined. The average energy given a $Q$ value is:

$$\langle E \rangle_Q = \frac{1}{R} \sum_{i=1}^{R} \frac{1}{\mathcal{N}_i(Q)} \sum_{j=1}^{\mathcal{N}_i(Q)} E_j^{(i)}(Q) \ . \tag{3}$$

The weights $p_i$ can be derived by imposing the $W(Q)$ variance to be minimum, obtaining:

$$S(Q) = \ln \left[ \frac{\sum_{i=1}^{R} \mathcal{N}_i(Q)}{\sum_{j=1}^{R} n_j e^{-\beta \langle E \rangle_Q + f_j}} \right] \tag{4}$$

for the entropy $S(Q) = \exp[W(Q)]$. This is solved self-consistently with:

$$e^{-f_k} = \sum_Q e^{-\beta_k \langle E \rangle_Q + S(Q)} \ . \tag{5}$$

The self-consistent equations can be solved iteratively with initialization $f_j = 0$, $j \in [1, R]$ and $S(Q) = 0$, $Q \in [0, 1]$. Convergence is reached when both $S(Q)$ and $f_k$ differ, on average in their respective domains, from the previous step by less than $10^{-14}$.

The advantage of WHAM is the possibility to obtain the free energy profile as a function of the reaction coordinate $F(Q) = \langle E \rangle_Q - T S(Q)$, in particular for temperatures that have not been sampled. The folding temperature $T_f$ is defined through the peak of the specific heat $C_v(T) = (\langle E^2 \rangle_T - \langle E \rangle_T^2)/(T^2 N)$. In the latter, averages are evaluated using WHAM results and $N$ is the number of residues. Hence, $T_f = \arg\max_T C_v(T)$. For each protein, the final folding temperature is the mean of at least two heat capacity profiles, each of which comes from a WHAM calculation using 8 independent simulations at equilibrium. These run for $2 \cdot 10^9$ integration steps, or $8.3 \cdot 10^4$ MD steps. The first 10% of the simulations is discarded from histogram counting, to allow for thermalization to take place.

## Gaussian entanglement

The linking number $G$ between two closed oriented curves $\gamma_i = \{r_i\}$ and $\gamma_j = \{r_j\}$ in $\mathbb{R}^3$ can be defined through Gauss' double integrals [42]:

$$G(\gamma_i, \gamma_j) := \frac{1}{4\pi} \oint_{\gamma_i} \oint_{\gamma_j} \frac{r_i - r_j}{|r_i - r_j|^3} \cdot \left( dr_i \times dr_j \right) \ . \tag{6}$$

This number is an integer and a topological invariant. A generalization for discrete and open curves is the Gaussian entanglement $G'$ [37–39]. For a chain with $N$ monomers, $\gamma = \{r_k\}_{k=1}^{N}$, let $\gamma_i = \{r_i\}_{i=i_1}^{i_2}$ and $\gamma_j = \{r_j\}_{j=j_1}^{j_2}$ be two non-overlapping portions of $\gamma$. We require $j_2 - j_1 \geq m_j$ and $i_2 - i_1 \geq m_i$. In the present work, we choose $m_i = 4$ and $m_j = 10$ [39]. In coarse-grained protein chains, $r_k$ represents the alpha carbon position vector. Let $R_i = (r_{i+1} + r_i)/2$ be the mean position between two subsequent alpha carbons and $\Delta R_i = r_{i+1} - r_i$ be the virtual bond vector. The Gaussian entanglement, actually describing the self-entanglement between two different

portions of the same chain, is defined as:

$$G'(\gamma_i, \gamma_j) := \frac{1}{4\pi} \sum_{i=i_1}^{i_2-1} \sum_{j=j_1}^{j_2-1} \frac{\boldsymbol{R}_i - \boldsymbol{R}_j}{|\boldsymbol{R}_i - \boldsymbol{R}_j|^3} \cdot \left( \Delta \boldsymbol{R}_i \times \Delta \boldsymbol{R}_j \right) \tag{7}$$

Crucially, $G'(\gamma_i, \gamma_j)$ is a real, not necessarily integer, number.

In the present contribution we consider $\gamma_i$ as a "loop", a chain portion "closed" by a non-covalent interaction, with residues $i_1$ and $i_2$ in contact with each other in the native configuration (see the Structure-based energy function subsection for details). No similar constraints are imposed on $\gamma_j$, which we call a "thread", due to its possible intertwining with a loop, if involved in an entangled motif.

For any loop $\gamma_i$, or equivalently for any native contact, we select the thread most likely entangling with $\gamma_i$ by maximizing $|G'(\gamma_i, \gamma_j)|$ over all threads $\gamma_j$ which do not overlap with $\gamma_i$. This yields also the Gaussian entanglement $G'$ (where the argument $\gamma_i$ has been dropped for clarity) associated to that loop, in combination with the selected thread, written formally as:

$$G' = \arg\max_{G'(\gamma_i, \gamma_j) \in \Omega(\gamma_i)} |G'(\gamma_i, \gamma_j)| \tag{8}$$

where $\Omega(\gamma_i)$ is the set of $G'(\gamma_i, \gamma_j)$ computed for all threads $\gamma_j$ that do not overlap with $\gamma_i$ and satisfy $j_2 - j_1 \geq m_j$.

An entanglement indicator for the whole protein configuration, given the distribution of Gaussian entanglement values selected for each loop with the latter strategy, can be defined in many different ways. The Gaussian entanglement that is maximum in modulus, and the corresponding loop-thread pair, were previously considered [39]. In the present contribution, we introduce as an entanglement indicator, for the whole chain configuration, the weighted average of $G'$ over all loops:

$$\langle G' \rangle = \frac{1}{H} \sum_{\gamma_i} G' \, h(|G'||g_0, m) \ . \tag{9}$$

The unnormalized weights are defined through an activation Hill function:

$$h(|G'||g_0, m) = \frac{1}{1 + \left( \dfrac{g_0}{|G'|} \right)^m} \tag{10}$$

where $g_0$ is the activation threshold, such that $h(g_0|g_0, m) = 1/2$, and $m$ is the cooperativity index. The larger $m$ the sharper the activation across the threshold. The normalization constant is the sum of all weights $H = \sum_{\gamma_i} h(|G'||g_0, m)$. During an MD simulation, the set of loops $\gamma_i$ that define the weighted average, and therefore the normalization $H$, correspond to the native contacts formed in the snapshot under consideration (see the Structure-based energy function subsection for details). We observe that the entanglement indicator $\langle G' \rangle$ is not overly sensitive, for example, to single contact formation/breaking events. Those might instead lead to abrupt changes in the maximum Gaussian entanglement $|G'|$, without being related to large-scale rearrangements of the whole chain.

The Gaussian entanglement for two closed curves is an integer number, thus in classifying the entanglement between two open curves we use $g_0 = 0.5$ as a threshold separating the $|G'| \approx 0$ case, for a loop which is not entangled with any other portion of the protein chain, from the $|G'| \approx 1$ case, for a loop which is instead threaded once by some other segment of the protein chain. Note that entangled loops had been previously defined in a much more restrictive way, by the $|G'| \geq 1$ constraint [39]. The entanglement indicator $\langle G' \rangle$ defined

here can be roughly interpreted as the average Gaussian entanglement for entangled loops ($|G'| > g_0$). However, the finite value of the cooperativity index, set in the present work to $m = 3$, makes for a less clear-cut separation close to the activation threshold, allowing $\langle G' \rangle$ to be used as a putative reaction coordinate for all values. In place of the Hill activation function used here, which outputs continuous values, a step function with the same activation threshold $g_0 = 0.5$, with discretized outputs, was instead used to estimate the difference in entanglement with respect to the native structure [48, 49, 69].

## Ensemble definition and pathway classification

From simulations at equilibrium, the Transition State Ensemble (TSE) of the SH3 domain is defined by means of the free energy profile as a function of the fraction of native contacts $F(Q)$ computed using WHAM. At the folding temperature $T_f$, the free energy profile has two degenerate minima, corresponding to the unfolded and folded states, separated by a free energy barrier located at $Q_{TSE} = 0.5$. A configuration belongs to the TSE if its fraction of native contacts falls into the range $[Q_{TSE} - \delta Q, Q_{TSE} + \delta Q]$, where $\delta Q = 0.03$.

Simulations used to study thermodynamic equilibrium start from the experimental PDB structures. For both proteins, the following method is used to select initial unfolded configurations of refolding events. For each protein, we pick uncorrelated configurations sampled from simulations at equilibrium above the folding temperature $T = cT_f > T_f$ ($c = 1.22$ and $c = 1.33$ for RD1 and SH3, respectively).

RD1 intermediate ensembles can be characterized only using both reaction coordinates $Q$ and $\langle G' \rangle$. Two-dimensional profiles are obtained from the Kernel Density Estimation (KDE) of the negative log-scale histogram of collected configurations. The KDE calculation uses the Python library `scipy` implementation [88] in which bandwidths are computed using Scott's rule [89] and rescaled by the data variances. Finally, contour plots, computed using the Python library `matplotlib` [90], visualize the KDE profiles. For all contour plots, each level corresponds to approximately 0.4 in log-histogram units. Those units correspond to $\kappa_B T$ for free energy profiles in the equilibrium case.

Both RD1's IT and IE− ensembles are identified by approximating the metastable minima in the refolding two-dimensional profile as rhomboidal areas, as shown in Fig G in S1 Appendix. IT corresponds to configurations with $Q \in [0.5, 0.75]$ and $-0.2Q < \langle G' \rangle < -0.2Q + 0.2$. Similarly, IE− configurations have $Q \in [0.32, 0.46]$ and $-1.07Q - 0.26 < \langle G' \rangle < -1.07Q - 0.11$.

Trajectories are classified into the pathways described in Table 1 considering a rolling average (temporal window of 555 MD Steps) of the reaction coordinates and, subsequently, testing the passage of the trajectories through the ensembles. States belonging to IT are defined through the rhomboidal area described above. F and U ensembles are defined by the sets $(Q, \langle G' \rangle) \in [0.75, 1] \cup [-1, -0.5]$ and $(Q, \langle G' \rangle) \in [0, 0.375] \cup [-0.25, 0.25]$, respectively. "Fast folding" trajectories refold from U to F without passing through IT. On the other hand, if the trajectories include IT states, they are classified using the last states before F, IT for "threading" and U for "backtracking". Examples of trajectories in the $(Q, \langle G' \rangle)$ phase space belonging to each of the four classes are shown in Fig H in S1 Appendix.

Contacts highlighted in red and magenta in Figs 6 and 8 are selected as follows: from each contact map probability for the IT and IE− ensembles, we compute the probability Z-scores as $z_i := (p_i - \langle p \rangle)/\sigma_p$, where $\langle p \rangle$ and $\sigma_p$ are the average and the standard deviation, respectively. Red framed contacts are defined as those having $z_i^{IE-} > 1$ in the IE− ensemble and $z_i^{IT} < 1$ in the IT ensemble. On the other hand, magenta contacts are selected among those contacts that satisfy the condition $z_i^{IE-} > 0$ and belong to the group of 30 entangled native contacts of the

RD1 protein. The list of which contacts belong to which groups is reported in Table A in S1 Appendix.

## Exponential fit of contact formation curves

To investigate the behaviour of each native contact during refolding kinetics, contact formation probabilities are studied. Given a native contact between residues $i$ and $j$, we define a binary variable $\sigma_{ij}(t)$ which is one if the contact is formed at time $t$, zero otherwise. A contact is formed if $r^{ij} < g\, r_0^{ij}$ with $g = 1.2$, consistently with $Q$ definition (see the Structure-based energy function subsection for details). Contact formation probabilities are defined as the averages $\langle \sigma_{ij} \rangle(t)$ over different refolding trajectories. All 100 realizations are considered for the SH3 domain, whereas only the 52 "fast" pathway ones are used for the RD1 protein. Contact formation probabilities usually start from low values typical of the unfolded state ensemble, and increase towards a plateau value $A$ when the native state is reached.

$\langle \sigma_{ij} \rangle(t)$ are affected by strong fluctuations, similar to what is observed in the $Q$ and $\langle G' \rangle$ time series. Therefore, to perform further analysis, we smooth these curves through a block average procedure. This is applied to the $Q(t)$ and $\langle G' \rangle(t)$ time series to estimate the folding time as well as to the contact formation probabilities $\langle \sigma_{ij} \rangle(t)$, with time windows of 250 and 100 MD steps, respectively.

After performing the block average, each $\langle \sigma_{ij} \rangle(t)$ curve is fit with an exponential function $A - B \exp(-kt)$ using nonlinear least squares implemented in the Python library `scipy` [88]. $A$ and $A - B$ estimate the contact formation probability in the F and the U ensemble, respectively. The contact folding rate $k$ is the kinetic observable representing the characteristic rate of transition from the unfolded to the native ensemble for the contact formation probability. Standard deviations are extracted from the minimization results [88].

## Supporting information

**S1 Appendix. This file includes one table (Table A) and eight figures (Fig A–Fig H).** (PDF)

**S1 File. This file includes the plots of the exponential fits to contact formation probability curves for all native contacts of the entangled RD1 protein.** Only the 52 trajectories refolding at $t = 0.9T_f$ through the fast channel are used to compute contact formation probabilities. All plots in this file are drawn as explained in the caption of Fig 7. (PDF)

**S2 File. This file includes the plots of the exponential fits to contact formation probability curves for all native contacts of the non-entangled SH3 domain.** All 100 refolding trajectories at $t = 0.9T_f$ are used to compute contact formation probabilities. All plots in this file are drawn as explained in the caption of Fig 7. (PDF)

**S1 Video. Fast folding event.** Folding transition along the fast pathway going through the short-lived intermediate IE$_-$. The color code is the same as in the upper left panel of Fig 1. (MP4)

**S2 Video. Threading event.** Threading transition along the slow pathway, from the kinetic trap IT to the native state F. The color code is the same as in the upper left panel of Fig 1. (MP4)

## Acknowledgments

We thank F. Seno for enlightening discussion. CloudVeneto is acknowledged for the use of computing and storage facilities.

## Author Contributions

**Conceptualization:** Marco Baiesi, Enzo Orlandini, Antonio Trovato.

**Data curation:** Leonardo Salicari.

**Funding acquisition:** Antonio Trovato.

**Investigation:** Leonardo Salicari, Antonio Trovato.

**Methodology:** Leonardo Salicari, Marco Baiesi, Enzo Orlandini, Antonio Trovato.

**Software:** Leonardo Salicari.

**Supervision:** Antonio Trovato.

**Writing – original draft:** Leonardo Salicari, Antonio Trovato.

**Writing – review & editing:** Leonardo Salicari, Marco Baiesi, Enzo Orlandini, Antonio Trovato.

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
