## [Decision Letter · Decision Letter 0]

19 Jul 2023

Dear Prof. Trovato,

Thank you very much for submitting your manuscript "Folding kinetics of an entangled protein" for consideration at PLOS Computational Biology.

As with all papers reviewed by the journal, your manuscript was reviewed by members of the editorial board and by several independent reviewers. The reviewers have overall found the manuscript interesting but have raised a number of concerns in terms of clarity of scientific presentation, contextualization, methodology, and analysis. We would be happy to consider a revised version that addresses the reviewers concerns and fulfills their suggestions.

We cannot make any decision about publication until we have seen the revised manuscript and your response to the reviewers' comments. Your revised manuscript will also be sent to reviewers for further evaluation.

[3] Minimize the number of self citations.

Sincerely,

Peter M Kasson

Academic Editor

PLOS Computational Biology

Arne Elofsson

Section Editor

PLOS Computational Biology

Reviewer's Responses to Questions

**Comments to the Authors:**

Reviewer #1: The manuscript describes computational studies on the folding pathways of two small proteins – a SH3 domain which is a classic two-state folder and which has a non-entangled native structure, and the antifreeze protein RD1 which has a natively entangled state. Simulations of the folding/unfolding behaviour of the two proteins are conducted at the folding temperature(Tf) under equilibrium conditions, as well as under folding conditions when T=0.9Tf. The authors find that the two proteins behave differently particularly with respect lower temperatures which favour folding. Whilst the SH3 domain behaves simply and does not populate entangled states under any conditions, the RD1 protein has more complex folding kinetics populating two distinct intermediate states – a short-lived one in which there are a few key contacts made which direct the protein directly to the natively entangled state via a fast-folding track, the other of which has much more native-like structure but is not entangled which is a much more long-lived intermediate which either has to partially unfold (backtrack) or undergo a threading event in order to reach the native entangled state. This forms the basis of a slow folding pathway.

Strengths

1. The paper is important for several different reasons. First, whilst there are now quite a few papers using simulations to study the folding of knotted and slip-knotted proteins, there are extremely few which focus on entangled but unknotted proteins. As there are many such examples in the protein structural databank, understanding the implications of entanglement on folding pathways is vital. Second, it has recently been shown by other groups, that near-native folding intermediates which have non-native entanglement, and thus, avoid the cellular quality control mechanisms are much more common than previously thought and may have significant effects on a cellular level.

2. Many suitable controls have been employed to ensure the robustness of the results presented, e.g., using different Lennard-Jones’ potentials, effects of temperature etc. Thus, this is a rigorous piece of research.

3. Introduction of a Gaussian entanglement indicator and using a weighted-average of this to determine whether a structure/state is entangled or not, is a very helpful way in which to analyse structures in addition to the routine number of native contacts. Clearly, combining the two as reaction coordinates provides more detail on the folding trajectories and is essential for studying these types of entangled structure.

Weaknesses (and points the authors should address when revising the manuscript).

1. Use of the term long-lived to describe the intermediate IT on the folding pathway of RD1. The authors may want to consider whether this is the most appropriate term (longer-lived may be better). This is a protein which both computationally and experimentally folds very fast. None of the intermediate states are very long-lived with respect to folding intermediates observed for many other proteins. Everything is relative, but this term may mislead some readers.

2. It is stated that the intermediate IE speeds up folding but no evidence is given to support this statement. The whole debate on whether intermediates speed up folding by reducing the conformational search or whether they slow down folding (by acting as kinetic traps), or whether they are neutral in terms of folding speed but still obligate species that are populated on folding pathways has been debated for more than 40 years. I would simply remove this sentence or rephrase it. Or, if the authors feel that they really have evidence to make the statement, then they should indicate what this is.

3. The paragraph linking folding pathway to function is completely speculative and unnecessary and could easily be removed.

4. The definitions of t* and k aren’t very clear. Further explicit description should be included, i.e., confusion over whether t* was the time to first make the contact or the lifetime of the contact.

5. The authors discuss the diffuse nature of the folding transition state for RD1. They may be interested and want to reference a study by Zhang & Jackson on the 5-2 knotted UCH-L1 which provided experimental evidence for the diffusive nature of the transition state on both folding pathways.

6. The authors relate the height of the energy barrier between denatured and native states as representing the cooperativity of folding, and therefore state that the folding of RD1 is more cooperative. This might be something which it is accepted in the computational field, but for experimentalists, the height of the barrier merely determines the rate at which the protein folds and doesn’t inform on cooperativity (which is associated with other partially folded states that might be populated on the folding pathway). Further clarification of how the term cooperativity is being used is required.

7. The term “unnecessarily hard” is used to describe the folding of knotted and entangled proteins in the Introduction. Although several decades ago when they were first identified it was assumed that the folding of knotted/entangled proteins would be very challenging, there is now substantial evidence that this is not always the case – experimental results have shown that for shallow knotted proteins the knot may only slow the folding rate by 3-10 fold, and considerably less than single mutations which do not affect the topology of the protein. In addition, experimental and computational studies have also established that in a cellular environment chaperones can accelerate the folding of knotted proteins by at least one, if not two, orders of magnitude. General statements on how knotted/entangled proteins fold are frequently misleading as they do not illustrate the diversity of behaviours that has now been found for topologically complex systems.

8. In general, the discussion is far too focussed on comparison of the results with those from other computational studies apart from the reference to the experiments on RD1. There is almost no discussion of any other experimental studies looking at the folding of knotted/entangled systems. This is a short-coming of the paper. For example, there is a reference to a computational study demonstrating that knotted topology forms in the early stages of folding. First, this is just a single computational study and other computational studies have addressed this question. Second, there is experimental evidence for some knotted proteins that the knot forms relatively late on the folding pathway, and isn’t present for example in intermediate states.

9. It would be most informative if some of the results could be related back to the sequence/structure of RD1, e.g., three contacts key for the fast folding track - it would be very interesting to know what these residues were in the protein. I assume that all native contacts are weighted the same by using a Go model. In reality, these three residues might form stronger contacts than average and may mean that a greater proportion of molecules fold via the fast track. One might expect these three residues to be more highly conserved than others – a quick sequence alignment study would demonstrate whether this is the case or not. (a note of caution about this though - for fast folding small non-entangled proteins there is only evidence that residues are conserved for overall stability not for folding rate, i.e, the residues involved in forming a critical folding nucleus are no more conserved than for those conserved for stability.

Reviewer #2: The review is uploaded as an attachment.

**Have the authors made all data and (if applicable) computational code underlying the findings in their manuscript fully available?**

Reviewer #1: Yes

Reviewer #2: Yes

PLOS authors have the option to publish the peer review history of their article (what does this mean?). If published, this will include your full peer review and any attached files.

Reviewer #1: No

Reviewer #2: No
---

## [Decision Letter · Decision Letter 1]

2 Nov 2023

Dear Prof. Trovato,

We are pleased to inform you that your manuscript 'Folding kinetics of an entangled protein' has been provisionally accepted for publication in PLOS Computational Biology.

Best regards,

Peter M Kasson

Academic Editor

PLOS Computational Biology

Arne Elofsson

Section Editor

PLOS Computational Biology

Reviewer's Responses to Questions

**Comments to the Authors:**

Reviewer #1: The authors have addressed all the comments made by the two reviewers and in doing so greatly improved the manuscript.

Reviewer #2: I thank the authors for addressing my comments and suggestions!

**Have the authors made all data and (if applicable) computational code underlying the findings in their manuscript fully available?**

Reviewer #1: Yes

Reviewer #2: None

PLOS authors have the option to publish the peer review history of their article (what does this mean?). If published, this will include your full peer review and any attached files.

Reviewer #1: No

Reviewer #2: No

---

## [Editor Report · Acceptance letter]

8 Nov 2023

PCOMPBIOL-D-23-00605R1 

Folding kinetics of an entangled protein

Dear Dr Trovato,

I am pleased to inform you that your manuscript has been formally accepted for publication in PLOS Computational Biology. Your manuscript is now with our production department and you will be notified of the publication date in due course.

With kind regards,

Zsofi Zombor
